# A Study of Isosorbide Synthesis from Sorbitol for Material Applications Using Isosorbide Dimethacrylate for Enhancement of Bio-Based Resins

**DOI:** 10.3390/polym15173640

**Published:** 2023-09-04

**Authors:** Vojtěch Jašek, Jan Fučík, Jiří Krhut, Ludmila Mravcova, Silvestr Figalla, Radek Přikryl

**Affiliations:** 1Institute of Materials Chemistry, Faculty of Chemistry, Brno University of Technology, 61200 Brno, Czech Republic; silvestr.figalla@vut.cz (S.F.); prikryl@fch.vut.cz (R.P.); 2Institute of Environmental Chemistry, Faculty of Chemistry, Brno University of Technology, 61200 Brno, Czech Republic; xcfucikj@vutbr.cz (J.F.); mravcova@fch.vut.cz (L.M.)

**Keywords:** isosorbide, isosorbide dimethacrylate, PHB, alkyl 3-hydroxybutyrates, enhancement, cross-linker

## Abstract

Bio-based cross-linkers can fulfill the role of enhancing additives in bio-sourced curable materials that do not compare with artificial resin precursors. Isosorbide dimethacrylate (ISDMMA) synthesized from isosorbide (ISD) can serve as a cross-linker from renewable sources. Isosorbide is a bicyclic carbon molecule produced by the reaction modification of sorbitol and the optimal conditions of this reaction were studied in this work. The reaction temperature of 130 °C and 1% *w*/*w* amount of *para*-toluenesulfonic acid (*p*-TSA) were determined as optimal and resulted in a yield of 81.9%. Isosorbide dimethacrylate was synthesized via nucleophilic substitution with methacrylic anhydride (MAA) with the conversion of 94.1% of anhydride. Formed ISD and ISDMMA were characterized via multiple verification methods (FT-IR, MS, ^1^H NMR, and XRD). Differential scanning calorimetry (DSC) proved the curability of ISDMMA (activation energy *E_a_* of 146.2 kJ/mol) and the heat-resistant index of ISDMMA (*T_s_* reaching value of 168.9) was determined using thermogravimetric analysis (TGA). Characterized ISDMMA was added to the precursor mixture containing methacrylated alkyl 3-hydroxybutyrates (methyl ester M3HBMMA and ethyl ester E3HBMMA), and the mixtures were cured via photo-initiation. The amount of ISDMMA cross-linker increased all measured parameters obtained via dynamic mechanical analysis (DMA), such as storage modulus (*E’*) and glass transition temperature (*T*_g_), and the calculated cross-linking densities (*ν_e_*). Therefore, the enhancement influence of bio-based ISDMMA on resins from renewable sources was confirmed.

## 1. Introduction

Isosorbide is a very promising molecule for a vast amount of different applications including medical [1,2], plasticizing [3,4,5], materials [6,7], or flame retardants [8,9,10]. This molecule is synthesized and manufactured from various starting substances. When sorbitol (a naturally occurring polyol) is used, a dehydration reaction is required to obtain isosorbide [11,12]. Many synthesis methods, including various catalysts, were introduced, experimentally assessed, and described in the published literature [13,14,15,16]. In addition to using sorbitol as a starting material for isosorbide synthesis, various carbohydrate structures, such as glucose [17] or starch [18,19,20], are used to manufacture this compound. However, a different approach needs to be used in such cases to obtain high yields of isosorbide. The mentioned starting materials are compounds containing carbonyl functional groups, meaning that they need to be reduced to alcohol (the alcohol formed from the reduced C6 carbonyl structure is sorbitol) [21,22]. After the reduction of carbonyl compounds, the dehydration process occurs, eventually forming isosorbide.

Isosorbide dimethacrylate, a diester of isosorbide and methacrylic acid, has a major potential in material applications [23,24,25]. This compound is synthesized via various reaction approaches such as a direct Fisher esterification involving isosorbide and methacrylic acid with an acidic catalyst [26]. Methacrylic anhydride can also be used as an appropriate nucleophile in the environment of a suitable base (such as N,N-dimethylaminopyridine (DMAP), or miscible carboxylates) [27,28,29]. Besides the acid and anhydride, methacryloyl chloride can serve as the most reactive type of nucleophile for this reaction [30]. Once isosorbide dimethacrylate is formed it can be used, especially as a cross-linking agent for numerous usages requiring curable resins [31]. It can be involved in 3D printing technologies (e.g., SLA) [32], as a component in bio-based coatings [33,34], or as a compound to enhance hardness, brittleness, and the glass transition temperature (*T*_g_) of prepared polymers and composites [35,36]. Furthermore, isosorbide dimethacrylate exhibits relatively low apparent viscosity values (tens to hundreds of mPa·s) which can be useful applications that require decreased viscosity while the retaining the final material properties [37].

This article shall be focused on the study of the synthesis of isosorbide from sorbitol using a commercially available acid (p-toluenesulfonic acid) as a catalyst via dehydration (intramolecular nucleophilic substitution). The appropriate conditions, such as the optimal reaction temperature or the effective amount of catalyst, will be investigated in order to accomplish the highest possible yield of formed isosorbide. Afterward, the synthesis of isosorbide dimethacrylate will be performed using methacrylic anhydride as a nucleophile involving the presence of an environmentally friendly catalyst—potassium acetate—which is an alternative to widely used DMAP as a catalyst. Numerous analytical methods (such as electrospray mass spectrometry (MS), nuclear magnetic resonance (NMR), and X-ray diffraction (XRD)) will be used for the structural verification of all synthesized products. The formed polymerizable diester of isosorbide will be characterized via differential scanning calorimetry (DSC) to prove the reactivity of the compound and its ability to be polymerized. In addition, thermal gravimetric analysis (TGA) will be performed to describe the heat-resistant index of the synthesized isosorbide dimethacrylate. Eventually, isosorbide dimethacrylate will be used to enhance the dynamic mechanical properties of previously synthesized polymerizable monoesters of 2-hydroxypropanoic acid and 3-hydroxybutanoic acid to describe the effect on the characteristics of the final resin.

## 2. Materials and Methods

### 2.1. Materials

Sorbitol (D-glucitol, 98%) used for the synthesis of isosorbide was purchased from Fichema s.r.o. (Brno, Czech Republic). The catalyst for the dehydration (*para*-toluenesulfonic acid (*p*-TSA), monohydrate, 98%, for synthesis) was acquired from Sigma-Aldrich (St. Louis, MO, USA). Methacrylic anhydride (94%, for synthesis) used for the modification of isosorbide to isosorbide dimethacrylate was obtained from Sigma-Aldrich (St. Louis, MO, USA). Other chemicals used in the described syntheses, namely potassium acetate (p.a.) as catalyst and sodium hydroxide (p.a.) as neutralizing agent, were obtained from PENTA s.r.o. (Prague, Czech Republic). Initiators for the curing experiments such as Luperox^®^ DI, *tert*-Butyl peroxide (for synthesis), or phenylbis(2,4,6-trimethylbenzoyl)phosphine oxide (BAPO, 97%) were purchased from Sigma-Aldrich (St. Louis, MO, USA).

### 2.2. Analytical Methods for Structural Characterization

#### 2.2.1. Fourier-Transform Infrared Spectroscopy (FT-IR)

Fourier-transform infrared spectroscopy served as a structural verification method. In the case of a particular synthesis of isosorbide dimethacrylate, it proved the disappearance of hydroxyl groups and the presence of ester bonding. Spectra were recorded by the infrared spectroscope Bruker Tensor 27 (Billerica, MA, USA) with attenuated total reflectance (ATR) (the dispersion component was a diamond). The irradiation source was an LED laser. The application of Michelson interferometer was required since Fourier transformation was performed. In total, 32 scans were obtained in 1 measurement with a measurement resolution of 2 cm^−1^.

#### 2.2.2. Mass Spectrometry (MS)

The spectrometer used for MS was a Bruker EVOQ LC-TQ (Billerica, MA, USA). Set MS conditions: ESI in positive; spray voltage 3500 V; cone temperature tempered at 340 °C; cone gas flow 30 a.u.; heated probe temperature 640 °C; probe gas flow 45 a.u., nebulizer gas flow 60 a.u.; and exhaust gas on. MRM transition for isosorbide (ISD): RT 0.78 min; 147.1 > 87.3 CE with 7.5 eV. Argon was used as a collision gas at a pressure of 1.4 mTorr. In addition, isosorbide dimethacrylate (ISDMMA) was verified by product scan; therefore, the mass spectrum of this compound was gained. Agilent Ion Trap 6320 LC/MS (Santa Clara, CA, USA) was set to the following working setup: nebulizer 172 kPa; drying gas flow 10 L/min; temperature of drying gas 350 °C; ionization mode ESI +; and complete scan used at a scan interval of 130–810 *m*/*z*.

#### 2.2.3. X-ray Diffraction Analysis (XRD)

XRD analyses were performed via X-ray instrumentation EMPYREAN (PANalytical, Malvern, UK) in a working setup with Bragg–Brentano parafocusing composition withCuKα radiation. The set conditions were: range 4–85° second, step resolution 0.015° second; voltage 40 kV; current applied 30 mA; ADS 10 mm; time per step 93 s; no monochromator.

#### 2.2.4. Nuclear Magnetic Resonance (NMR)

Nuclear magnetic resonance (NMR) was performed for the detailed structural verification of produced compounds. The measurements were obtained by a Bruker Avance III 500 MHz (Bruker Billerica, MA, USA) at the temperature of 30 °C using deuterium chloroform (CDCl_3_) as a dispersion continuum along with tetramethylsilane as an internal standardized compound (TMS) for these purposes. The acquisition time was 4.0 s. The chemical shift (*δ*) units are part per million (ppm) with reference to TMS. Coupling constants *J* (with a unit of (Hz)) were described as follows: s for the singlet, d for the doublet, t for the triplet, q for the quartet, p for the quintet, m for the multiplet.

### 2.3. Synthesis of Isosorbide form Sorbitol

Sorbitol (400 g, 2.2 mol) was added into a 1 L jacket reactor homogenized with a stirrer (500 rpm) and the reactor was heated up to the reaction temperature (120–140 °C). A vacuum pump was applied to the apparatus and the pressure was decreased to approximately 1–2 kPa to separate adsorbed water from sorbitol. The particular amount of catalyst (*p*-TSA, 0.25–1.00 mol.%) was added after 15 min of homogenization and the pressure decrease was set to 3.5–5 kPa. Reaction water was continuously distilled and condensed during the reaction. When the reaction time expired (5 h), the catalyst was neutralized with sodium hydroxide and then the product (isosorbide) was distilled from the batch. The conditions of the separation were 0.3–0.5 kPa and 180–200 °C. Distilled and condensed isosorbide was left to spontaneously crystallize and it was structurally characterized afterwards. Reaction schemes, including the dehydration of sorbitol to isosorbide (including sorbitan as an intermediate), are displayed in Figure 1 and Figure 2.

### 2.4. Synthesis of Isosorbide Dimethacrylate

Previously synthesized and characterized isosorbide (200 g, 1.4 mol) was poured into a round-bottomed flask that was transferred into an oil bath for temperature regulation. The temperature was set at 70 °C and a magnetic stirrer was applied. Once isosorbide reached the required temperature, methacrylic anhydride (432 g, 2.8 mol) was added into the flask and the temperature of the mixture was increased to 70 °C again. Then, the catalyst was added to the mixture (potassium acetate, 6.9 g, 0.014 mol) and the reaction was performed for 8 h and monitored via GC-FID analysis. The reaction mixture was purified of the formed methacrylic acid by neutralization by sodium hydroxide and formed methacrylate salts were extracted by distilled water. Eventually, neutralized isosorbide dimethacrylate was dried with sodium sulfate, filtered, and structurally characterized. The reaction mechanism is displayed in Figure 3.

### 2.5. Analytical Methods for Kinetics Study of Syntheses

#### 2.5.1. Liquid Chromatography with Mass Spectrometry (LC-MS)

Samples of isosorbide gained during a continuous reaction were analyzed by UHPLC Agilent 1290 Infinity LC (Santa Clara, CA, USA) connected with triple quadrupole (Bruker EVOQ LC-TQ) (Billerica, MA, USA) equipped with atmospheric ESI. An external source of nitrogen and air (generator of gases: Peak Scientific—Genius 3045) (Glasgow, Scotland) was used. Luna^®^ Omega Polar C18 Phenomenex (100 × 2.1 mm, 1.6 µm) (Torrance, CA, USA) was the stationary phase. The working column temperature was set to 40 °C while the flow rate reached 0.5 mL/min. Mobile phases responsible for the separation in LC were (A) 0.1% HCOOH in MilliQ water and (B) acetonitrile, and used in the following conditions: An eluent (%): t(0 min) = 85, t(0.6 min) = 80, t(3.7 min) = 5, t(4.8 min) = 90. Analyses were stopped after 6.0 min, and re-equilibration took 2.0 min. The injection volume was 7 µL. Mass spectrometry parameters were described earlier (see Section 2.2.2).

#### 2.5.2. Gas Chromatography with Flame Ionization Detector (GC-FID)

Samples were obtained during the methacrylation of isosorbide, and quantification of reactants (methacrylic anhydride and methacrylic acid) was performedvia GC-FID Hewlett Packard 5890 Series II (Palo Alto, CA, USA). Nitrogen (as an auxiliary gas), air (as an oxidizer), and hydrogen (as a carrier) were applied to the instrumentation for measurements. The stationary phase was ZB-624 (60 m × 0.32 mm, 1.8 µm). Inlet temperature: 210 °C; detector temperature: 250 °C. The temperature gradient was applied, the initial temperature was 60 °C (1-min maintenance) then at emperature rate of 20 °C/min was used and the eventual temperature was 260 °C (tempered for 15 min). The flow of the analyzed mixture was 3 mL/min with a 1:40 split ratio. Injection volume: 1 µL. Retention times of measured compounds: methacrylic anhydride (MAA) (RT9.88 min); methacrylic acid (MA) (RT 6.05 min).

### 2.6. Reactivity and Thermal Stability Characterization of Isosorbide Dimethacrylate

#### 2.6.1. Differential Scanning Calorimetry (DSC)

DSC was used to describe the reactivity kinetics of produced polymerizable isosorbide dimethacrylate. The product was mixed with a thermo-initiator, namely Luperox^®^ DI (1% *w*/*w* amount). Aluminum pans (10–15 mg) were filled with the mixtures and were hermetically sealed. Instrument DSC 2500 model (TA Instruments) (New Castle, DE, USA) was the instrumentation used for measurements. Four different heating rates were performed on each sample from temperatures of 10 to 245 °C while the continual temperature increases were: 5; 10; 15 and 20 °C/min. One measurement cycle was performed for each sample since the initiator reacted during one cycle. An inert atmosphere (nitrogen) was applied during all measurements.

#### 2.6.2. Thermogravimetric Analysis (TGA)

TGA was used to acquire the heat stability index of polymerized isosorbide dimethacrylate. Used samples were obtained similarly to the samples for FT-IR curability analysis. ISDMMA was mixed with BAPO (1% *w/w*), polymerized by the photoinitiator for 30 min using an LED irradiation source of 405 nm wavelength. The instrumentation used was a TGA Q500 from (TA Instruments) (New Castle, DE, USA). Analyzed samples (around 15 mg) were measured in the following conditions: equilibration at 35 °C; temperature increase to 600 °C with applied temperature ramp of 10 °C/min under nitrogen; 15 min at 600 °C under an oxidizing environment.

### 2.7. Thermo-Mechanical Characterization of Synthesized Isosorbide Dimethacrylate Containing Resins

#### Dynamic Mechanical Analysis (DMA)

The thermo-mechanical properties of mixtures containing isosorbide dimethacrylate with resin precursors based on methacrylated 3-hydroxybutyrates (see Figure 1) were measured with DMA 2980 from (TA Instruments, New Castle, DE, USA). The measured objects were prepared from isosorbide dimethacrylate and methacrylated alkyl 3-hydroxybutyrate with a specified amount of ISDMMA in the mixture (0–10% *w/w* of ISDMMA). Then, 1% *w/w* of BAPO was added and precursors were polymerized for 30 min (405 nm LED). Tested objects had parameters: 60 × 10 × 2 mm. Objects were applied into a dual cantilever attachment and the parameters of applied deformation were: 10 μm amplitude, 1 Hz frequency. The temperature increase was from 25 °C to 120 °C with an increase rate of 3 °C/min temperature.

## 3. Results and Discussion

### 3.1. Isosorbide Synthesis and Characterization

The monitoring of isosorbide synthesis was realized via the weighing of the condensed reaction water as a very quick a simple method to determine the most appropriate reaction temperature. The results of the collected reaction water formed during the dehydration are shown in Figure 1a. Once the most optimal temperature was determined, the reaction kinetics monitoring next focused on the differing amount of the catalyst in the reaction mixture. These reactions were monitored more precisely using LC-MS as a quantification method for isosorbide as a forming product. This study was performed to obtain information regarding the differences in the rate of reaction based on the quantity of the catalyst. The results of these measurements are displayed in Figure 1b.

**Figure 1 polymers-15-03640-f001:**
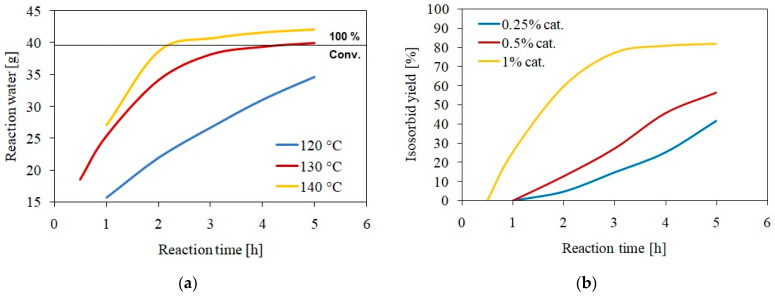
(**a**) The dependence of formed and condensed reaction water during the dehydration reaction resulting in the production of isosorbide on the reaction time at different temperatures; (**b**) Monitoring of the synthesis of isosorbide via LC–MS analysis for differing amounts of catalyst (p-TSA) at 130 °C.

The results showed that the most appropriate reaction temperature was 130 °C as the equilibrium at this temperature was reached while almost the exact theoretical amount of condensed reaction water (101.5%) was collected. At the temperature of 140 °C, more than the theoretical quantity of water was collected, which indicated the occurrence of side reactions forming degradation products. The degradation products that cannot be further dehydrated forming isosorbide are 1,5-sorbitan and 2,5-sorbitan [11,13]. These compounds are also formed and the reaction water is generated during this process. However, a temperature of 120 °C exhibited a much lower reaction rate than 130 °C. From these results, the dependence of isosorbide analyzed yield (via LC-MS) on time was described at the reaction temperature of 130 °C. LC-MS analysis revealed that 1% *w*/*w* of the used catalyst (*p*-TSA) led to the highest yield of formed isosorbide (81.9%). Lower amounts of catalyst led to significantly lower yields compared to 1% *w*/*w* catalyst. In particular, 0.5% *w*/*w* catalyst resulted in a yield value of 56.4% and 0.25% *w*/*w* catalyst reached 41.5% of isosorbide yield.

Structural verification via the FT-IR method is shown in Figure 2. The spectrum is a characteristic fingerprint of the compound. The signal at approximately 3600–3100 cm^−1^ is extremely intense as the isosorbide molecule contains two hydroxyl groups that are significant for the further modification to isosorbide dimethacrylate. This method was also used to confirm the presence of modifiable hydroxyl groups in the structure. This IR spectrum of isosorbide can be compared to published spectra in literature [38].

Mass spectrometry confirmed the structure of isosorbide and the spectrum is shown in Figure 3. The fragmentation of isosorbide corresponds with the prediction; the molecular ion peak obtained via mass spectrometry and present in the spectrum has the exact value of *m*/*z* (147.2 *m*/*z*).

X-ray diffraction analysis also verified the structure of the synthesized product. This method was used since isosorbide has a crystalline form. The XRD spectrum with the particular values of 2θ is presented in Figure 4. This spectrum also serves as a fingerprint of the molecule. Furthermore, the XRD spectrum of isosorbide was already published and is comparable with that obtained in this article [39].

Structural verification was also provided via the ^1^H NMR method and the final spectrum is shown in Figure 5. All peaks with appropriate chemical shifts have the same positions as the prediction. Shifts and coupling constants are described in Figure 5.

### 3.2. Isosorbide Dimethacrylate Synthesis and Characterization

The synthesis of isosorbide dimethacrylate involved a reaction with methacrylic anhydride (MAA) (catalysis of potassium acetate) and the products of this particular nucleophilic substitution are the ester of isosorbide and the secondary product—methacrylic acid (MA) (see Figure 3). The principle chosen for reaction kinetics monitoring was to quantify the decreasing reactant (MAA) and at the same time quantify the formation of methacrylic acid (MA). Since these two substances can be vaporized, GC-FID was used for the quantification. The reaction mixture consisted of an equimolar amount of MAA to isosorbide; therefore, equilibrium heading to the minimization of MAA in the mixture was awaited. The results of GC-FID quantification of the mentioned compounds are displayed in Figure 6.

It is evident that the majority of methacrylic anhydride reacted (94.1%) and that equilibrium tended to occur after 8 h of reaction. Simultaneously, the conversion of the forming methacrylic acid (95.1%) corresponded with the decrease in MAA. After purification and drying, isosorbide dimethacrylate was harvested resulting, in an ISDMMA yield of 71% of the theory. This decrease in product yield compared to the theoretical yield can be caused by the minor formation of isosorbide monomethacrylate, which is water-soluble and could be extracted from the mixture during the purification process.

The FT-IR spectrum of synthesized isosorbide dimethacrylate is displayed in Figure 7. The hydroxyl region (3600–3100 cm^−1^) completely disappeared, which confirmed the modification via esterification. The presence of ester bonding is proved by signals at wavenumber intervals of 1750–1735 cm^−1^ (referring to C=O stretching) and 1210–1150 cm^−1^ (belonging to C-O stretching). Since methacrylic functional groups contain unsaturated double bonds, those could be verified via signals at 1670–1600 cm^−1^ (C=C stretching) and 970–930 cm^−1^ and 850–800 cm^−1^ (C=C bending).

Mass spectrometry verified the structure of isosorbide dimethacrylate and the spectrum is shown in Figure 8. The full-scan spectrum of isosorbide dimethacrylate corresponds to the prediction and molecular peak obtained via mass spectrometry and present in the spectrum has the exact value of *m*/*z* (282.4 *m*/*z*). In addition, the spectrum contains a molecular peak with the Na^+^ cation as an adduct (304.4 *m*/*z*).

Structural confirmation was provided via the ^1^H NMR method and the final spectrum is shown in Figure 9. All peaks with appropriate chemical shifts have the same positions as the prediction. Shifts and coupling constants are described in Figure 9.

### 3.3. Curability and Thermostability of Synthesized Isosorbide Dimethacrylate

Synthesized isosorbide dimethacrylate was analyzed by differential scanning calorimetry (DSC) to obtain data regarding polymerizability. The process included the addition of the thermal initiator (Luperox^®^ DI) into the samples and the subsequent monitoring of the dependence of heat flow on increasing temperature while the temperature ramps varied. This procedure was performed to gain parameters Kissinger’s theory. According to this theory, the heating rate (*β*) and obtained temperatures of exothermal peaks (*T_p_*) are accessible from the following equation [40]:(1)$$\ln\left(\frac{\beta }{{T_{p}}^{2}}\right)=\ln\left(\frac{AR}{E}\right)-\frac{E}{R}\cdot \frac{1}{T_{p}},$$
where *β* refers to the heating rate (°C/min) and *T_p_* is the particular temperature of a maximum exothermic peak (°C). Kinetic parameters calculated from the equation are: *A* describes the pre-exponential factor (−); *E* is the activation energy (J/mol), and *R* is the gas constant (J/(mol·K)). The obtained results of each measurement of a particular temperature ramp are shown in Figure 10 along with the graphical representation of Kissinger’s theory for ISDMMA. The calculated kinetics parameters for the synthesized product are shown in Table 1.

The results of the DSC analysis confirm the polymerization activity of synthesized isosorbide dimethacrylate (ISDMMA). Moreover, quantities such as activation energy (*E*) and pre-exponential factor (*A*) were calculated using Kissinger’s equation. The results correspond with previously obtained kinetics study of isosorbide dimethacrylate [41].

In addition, TGA was used to determine the thermal stability of cured ISDMMA harvested after the DSC. This characterization is presented via a calculated heat-resistant index (*T_s_*), which evaluates the dependence of weight loss on increasing temperature and serves as a comparing parameter for different heat-stable materials. The mathematical formula used for the calculation has been reported in the published literature [42,43] and it is represented in Equation (2):(2)$$T_{s}=0.49\;[T_{5}+0.6(T_{30}-T_{5})],$$
where *T_s_* stands as the heat-resistant index (−). Specific temperatures obtained from TGA and used for the calculations were: *T*_5_ as the temperature at 5% of mass loss (°C) and *T*_30_ as the temperature at 30% of mass loss (°C). The TGA curves obtained from the analysis are shown in Figure 11 and the calculated heat-resistant index of cured ISDMMA is evaluated in Table 2.

The DSC and TGA analyses revealed that ISDMMA is a polymerizable monomer that can be used for cross-linking applications due to the nature of its molecular structure, as reported in previously published articles [44,45]. Moreover, its heat-resistant index reaches a value appropriate in various usage fields.

### 3.4. Ability of Isosorbide Dimethacrylate to Enhance Thermo-Mechanical Properties

The dynamic thermo-mechanical properties of cured thermosets containing isosorbide dimethacrylate (ISDMMA) and methacrylated monoesters of 3-hydroxybutanoic acid—methacrylated methyl 3-hydroxybutyrate (M3HBMMA) and methacrylated ethyl 3-hydroxybutyrate (E3HBMMA) (see Figure 12)—were studied via DMA analysis. ISDMMA was used as a cross-linker because of its molecular structure, reactivity, and properties. Alkyl 3-hydroxybutyrates are promising upcycled precursors synthesized from poly(3-hydroxybutyrate) (PHB) that can serve as reactants produced from low-molecular-weight polymer waste (low potential manufacture waste). The syntheses of methacrylated alkyl 3-hydroxybutyrates were described in our previous article [46].

Synthesized polymerized mixtures of ISDMMA and methacrylated alkyl esters were analyzed to obtain their storage modulus (*E’*), the damping factor (tan *δ*), the glass transition temperature (*T*_g_) (*T*_g_ was calculated from loss factor as its maximum), and the cross-linking density (*ν_e_*), which is the property of the thermoset especially important for enhanced cross-linked resins and is accessible from following equation [47]:(3)$$\upsilon_{e}=\frac{E^{\prime }}{3RT\prime },$$
where *ν_e_* is the cross-linking density (mol/m^3^); *E’* is the storage modulus (which the sample has in the rubbery plateau region) (E’ at *T*_g_ + 40 °C) (Pa); *T’* is the particular thermodynamic temperature (at *T*_g_ + 40 °C) (K), and *R* is the gas constant (J/(mol·K)). The DMA curves describing the dependence of the storage modulus of resins on temperature and graphs containing data regarding the damping modulus are shown in Figure 13 and Figure 14. Furthermore, the comparable values of storage modulus at different temperatures, the evaluated glass transition temperatures of particular systems, and the calculated cross-linking densities of cured resins are summarized in Table 3.

The DMA analysis of prepared mixtures containing both methacrylated alkyl 3-hydroxybutyrates and cross-linking agent—synthesized ISDMMA—confirmed the hardening role of ISDMMA. The comparison of the storage modulus of both raw and cross-linked cured monoesters revealed that all values of *E’* of resins consisting of M3HBMMA with the cross-linker are approximately twice as high as the values of systems with E3HBMMA. This result indicates that M3HBMMA as a curable precursor is more appropriate for high-performace applications. The values of *E’* at 30 °C were chosen for comparison purposes due to the fact that this parameter is often mentioned for similar materials and gives an idea of the application opportunities. Several materials’ values can be found in the literature [48,49]. Moreover, the glass transition temperature of both systems without a sufficient amount of cross-linker also reaches considerably higher values for systems with M3HBMMA. These results are probably caused by the better polymeric network formed by the methyl ester than by the ethyl ester of 3-hydroxybutanoic acid. These two curable precursors have not been characterized in literature yet; however, similar trends can be found with varying glass transition temperatures regarding the length of molecular structure in published articles [50,51]. The enhancing influence of the presence of ISDMMA on the cured resins is evident from the increase in the glass transition temperatures of both systems and also from the increase in the cross-linking densities calculated for all cured materials. In addition, the storage modulus increases when ISDMMA is added.

## 4. Conclusions

This work was focused on the study of the synthesis of isosorbide that served as a reactant for the preparation of a cross-linking agent—isosorbide dimethacrylate—used for the study of a potential mechanical enhancement of curable bio-based precursors based on depolymerized and modified poly(3-hydroxybutyrate). The optimal reaction conditions for isosorbide synthesis were 130 °C and 1% *w*/*w* amount of catalyst (*para*-toluenesulfonic acid). With these conditions, the yield of isosorbide was 81.9%. The synthesized product was modified via nucleophilic substitution using methacrylic anhydride (conversion of anhydride 94.1%) and a yield of 71.0% of formed isosorbide dimethacrylate was harvested. The curability and thermal stability of isosorbide dimethacrylate were measured resulting in the calculated activation energy of 146.2 kJ/mol and the heat-resistant index of 168.9. The enhancement influence of isosorbide dimethacrylate on the thermo-mechanical properties of cured systems based on methacrylated alkyl 3-hydroxybutyrates was confirmed. The storage modulus of all systems increased with the amount of cross-linking agent (68.6% increase for methyl ester and 123.8% for ethyl ester when 10% *w*/*w* of cross-linker was added). In addition, the presence of isosorbide dimethacrylate increased the glass transition temperature and cross-linking densities of all systems. Therefore, the enhancement of thermo-mechanical properties using novel bio-based curable systems based on depolymerized poly(3-hydroxybutyrate) with the addition of isosorbide dimethacrylate was confirmed.

## Data Availability

Not applicable.

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
