# Peer review of "A Study of Isosorbide Synthesis from Sorbitol for Material Applications Using Isosorbide Dimethacrylate for Enhancement of Bio-Based Resins"

_polymers, 2023, doi:10.3390/polym15173640_

Round 1

Reviewer 1 Report

Comments on the reviewed article:

1.       General remark. The data of suppliers of chemical raw materials and equipment manufacturers should be described in more detail in accordance with the requirements of the Polymers journal.

2.       Subsection 2.2.1. Authors used words: Spectroscopy and Spectrometry. Please use only the word spectroscopy consistently.

3.       What was the acquisition time during the NMR study? The information must be added to the text of the manuscript.

4.       Why did the Authors perform only 1H NMR and not also 13C NMR? This study would also be very valuable for this article.

5.       Schemes 1 and 2 are not fully legible. I suggest rearranging and enlarging them for sharpening. p-TSA structures are smeared.

6.       DSC analysis. I understand that the authors meant measurement at different heating rates, not heating scans. In how many measurement cycles was one sample tested at a given heating rate. In what atomosphere was the measurement carried out - oxidizing or inert? This information is missing in the text.

7.       Figure 1. The authors write about side reactions that occur in the process and lead to overestimation of yield. Please indicate the specific processes that take place (with the release of water). Determining that degradation processes occur is too general.

8.       Figure 2. Bonds assigned to given spectral peaks should be plotted on the figure.

9.       Figure 5. Looks like it's been scanned. Authors should subject it to graphic processing. Sharpen and enlarge the scale. I also suggest deleting the items marked in blue and the "measurement file name".

10.   Figure 7. Analogous remark as for Figure 2.

11.   Figure 11 also looks like a scan. The figure must be provided in the appropriate quality.

12.   Authors should check the article for linguistic correctness, because the article requires minor language corrections.

Authors should check the article for linguistic correctness, because the article requires minor language corrections.

Reviewer 2 Report

This paper described the synthesis of isosorbide dimethacrylate and its use as bio-based cross-linker. The dehydration reaction to produce isosorbide was followed by water evolution and LC-MS analysis. The methacrlation reaction was followed by the consumption of methacrylic anhydride and production of methacrylic acid. All product was also verified by FTIR, MS, and 1H NMR. XRD was used to confirm the crystalline structure of isosorbide. DSC was used to derive activation energy of the thermal curing process, TGA was used to obtain heat-resistant index. DMA was used to evaluate the suitability of the synthesized isosorbide dimethacrylate as crosslinker in bio-based methacrylated resin. While the scientific descriptions are adequate, careful proofreading is needed before acceptance.

Including but not limited to the following:

In abstract, “systems were photo-initially polymerized” needs revising, could be “the polymerization systems were photo-initiated.”

Page 2, “elecrospray” to “electrospray”

Page 4, “the mixture was tempered to”  to “the temperature of the mixture was increased to”

Page 8, second to last line, “was ale used” to “was also used”

Page 11, NMR resonance peak info was duplicated above Figure 9. Delete it.

Page 13, “has been multiply reported” , delete “multiply”.

Equation 2, delete Celsius sign in the subscript, it is confusing and unnecessary, do the same in the following paragraph.

Page 14, the line after equation (3), “saple” to “sample”

Table 3, explain why E’ at 30 C was used for comparison, reference it in the paragraph below.

Some of the English usages need revising. Generally OK and understandable. 

Round 2

Reviewer 1 Report

Article can be published in present form.